# Inhibition of Renal Stellate Cell Activation Reduces Renal Fibrosis

**DOI:** 10.3390/biomedicines8100431

**Published:** 2020-10-19

**Authors:** Jin Joo Cha, Chanchal Mandal, Jung Yeon Ghee, Ji Ae Yoo, Mi Jin Lee, Young Sun Kang, Young Youl Hyun, Ji Eun Lee, Hyun Wook Kim, Sang Youb Han, Jee Young Han, Ah Young Chung, Dae Wui Yoon, Im Joo Rhyu, Junseo Oh, Dae Ryong Cha

**Affiliations:** 1Department of Nephrology, Korea University Ansan Hospital, Ansan 15355, Korea; pearl0827@naver.com (J.J.C.); gheejy@nate.com (J.Y.G.); ji1052@hanmail.net (J.A.Y.); bluemarine2001@hanmail.net (M.J.L.); starch70@korea.ac.kr (Y.S.K.); 2Department of Anatomy, Korea University College of Medicine, Seoul 02841, Korea; chanchalbge@gmail.com (C.M.); dkdud0229@naver.com (A.Y.C.); ydw@jwu.ac.kr (D.W.Y.); irhyu@korea.ac.kr (I.J.R.); 3Department of Nephrology, Kangbuk Samsung Hospital, Sungkyunkwan University, Seoul 03181, Korea; femur0@naver.com; 4Department of Nephrology, Wonkwang University Sanbon Hospital, Gunpo 15865, Korea; borisoo12@hanmail.net (J.E.L.); bluejayway@naver.com (H.W.K.); 5Department of Nephrology, Inje University Ilsan Paik Hospital, Ilsan 10380, Korea; hansy1012@gmail.com; 6Department of Pathology, Inha University Hospital, Incheon 22332, Korea; jeeyhan@inha.ac.kr

**Keywords:** renal fibrosis, stellate cell, myofibroblast, albumin

## Abstract

Interstitial fibrosis is a common feature of chronic kidney disease, and platelet-derived growth factor receptor-β (PDGFR-β)-positive mesenchymal cells are reportedly the major source of scar-producing myofibroblasts. We had previously demonstrated that albumin and its derivative R-III (a retinol-binding protein-albumin domain III fusion protein) inhibited the transdifferentiation/activation of hepatic stellate cells (HSCs) to myofibroblasts and that R-III administration reduced liver fibrosis. In this study, we isolated cells (referred to as renal stellate cells, RSCs) from rat kidney tissues using the HSC isolation protocol and compared their morphological and biochemical characteristics with those of HSCs. RSCs shared many characteristics with HSCs, such as storage of vitamin A-containing lipid droplets and expression of HSC markers as well as pericyte markers. RSCs underwent spontaneous transdifferentiation into myofibroblasts in in vitro culture, which was inhibited by albumin expression or R-III treatment. We also evaluated the therapeutic effects of R-III in unilateral ureteral obstruction (UUO)-induced renal fibrosis in mice. Injected R-III localized predominantly in cytoglobin/stellate cell activation-associated protein (Cygb/STAP)-positive cells in the kidney and reduced renal fibrosis. These findings suggest that RSCs can be recognized as the renal counterparts of HSCs and that RSCs represent an attractive therapeutic target for anti-fibrotic therapy.

## 1. Introduction

Chronic kidney disease (CKD) represents a global health burden that affects >10% of adults worldwide, and renal fibrosis, particularly tubulointerstitial fibrosis, is the common final outcome of almost all progressive CKD [1]. Renal tubulointerstitial fibrosis is characterized by loss of tubular architecture with extracellular matrix (ECM) accumulation and tubular atrophy, resulting in organ dysfunction and kidney failure [2]. The traditional view is that myofibroblasts, responsible for excessive ECM deposition, are derived from resident fibroblasts, which are spindle-shaped cells of mesenchymal origin, but this process is poorly understood [3]. Several other candidate cells have also been reported as potential sources of myofibroblasts. Until a decade ago, it was hypothesized that the transition of tubular epithelial cells to a mesenchymal phenotype (EMT) is responsible for the generation of myofibroblasts [4]. The potential role of circulating fibrocytes from the bone marrow in the generation of myofibroblasts has also been described, but their role in renal fibrosis is not universally acknowledged [5]. Recent studies using genetic lineage mapping studies suggested that pericytes and perivascular fibroblasts are the major source of scar-producing myofibroblasts following kidney injury [6,7]. However, despite extensive research, except for the removal of the causative agent and organ transplantation there is no approved therapy for specifically targeting tissue fibrosis as of yet.

Liver pericytes, commonly known as hepatic stellate cells (HSCs), are found in the perisinusoidal space of the liver and constitute approximately 5–8% of total liver cells [8]. They are present in a quiescent state, exhibit non-proliferative characteristics in the normal liver, and store approximately 80% of the total vitamin A (retinol) present in the body as retinyl esters within the cytoplasmic lipid droplets [9]. In response to fibrogenic stimuli, quiescent HSCs undergo functional and phenotypic changes, referred to as “activation,” and transdifferentiate into myofibroblast-like cells [10]. This process is characterized by the loss of vitamin A-containing cytoplasmic lipid droplets, increased cellular proliferation, positive staining for alpha-smooth muscle actin (α-SMA), and enhanced synthesis of ECM proteins. It is well known that HSC activation plays a critical role in liver fibrosis, which is characterized by excessive deposition of ECM components [11]. Cells resembling HSCs were isolated from the pancreas in the late 1990s [12], and similar to HSCs, these pancreatic stellate cells (PSCs) were also found to play an important role in pancreatic fibrogenesis [13]. Thus, stellate cells (SCs) are considered an attractive target for anti-fibrotic therapies [14]. A previous study had suggested that cells with vitamin A-containing lipid droplets were also present in extrahepatic tissues, such as the lung, kidneys, and intestine [15].

Albumin, an abundant multifunctional plasma protein, is synthesized primarily by liver cells [16]. It is composed of three homologous domains (I, II, and III) and binds a wide variety of hydrophobic ligands, including fatty acids and retinoids [17,18]. In a previous study, we showed that albumin was endogenously expressed in quiescent SCs, but not in activated SCs, and that its forced expression in activated SCs induced a phenotypic reversion of myofibroblasts into early activated cell phenotype, accompanied with the reappearance of cytoplasmic lipid droplets and reduced expression of α-SMA and collagen type I [19]. On the basis of this finding, we developed a recombinant fusion protein (designated R-III) as an anti-fibrotic agent, in which the domain III of albumin was fused to the C-terminus of retinol-binding protein (RBP) [20]. We selected RBP for targeted SC delivery, as RBP and its membrane receptor (STRA6) coordinate the cellular uptake of retinol into HSCs [21]. Our follow-up study showed that R-III inhibited HSC activation in vitro and reduced liver fibrosis in vivo [22]. Hence, in the present study, we investigated whether cells resembling HSCs are present in kidney tissues and determined the therapeutic effects of R-III on unilateral ureteral obstruction (UUO)-induced renal fibrosis.

## 2. Experimental Section

### 2.1. Materials

Male Sprague-Dawley rats (six to eight weeks old) and male C57BL/6 mice (six to eight weeks old) were purchased from OrientBio, Inc. (Seongnam, Korea) and maintained under temperature-, humidity-, and light-controlled conditions. Animal experiments were approved by our institutional review board (KOREA-2018-0173) and complied with the Guide for the Care and Use of Laboratory Animals. The rat renal proximal tubular cell line NRK-52E and rat renal normal fibroblast cell line NRK-49F were obtained from the American Type Culture Collection (Rockville, MD, USA) and cultured in Dulbecco’s modified Eagle’s medium (DMEM) supplemented with 10% fetal bovine serum (FBS). The fusion protein R-III (Appendix A) was synthesized using a previously described method [20].

### 2.2. Isolation of Rat HSCs and RSCs and Cell Culture

HSCs were isolated from male Sprague-Dawley rats (nine weeks old) as described previously [19]. For the isolation of RSCs, the kidneys were perfused in situ with phosphate-buffered saline (PBS) and then with Hank’s buffered salt solution (HBSS) supplemented with collagenase, pronase (Sigma-Aldrich, St, Louis, MO, USA), and DNase (MP Biomedicals, Santa Ana, CA, USA) via the inferior vena cava. The perfused kidneys were dissected, and the outer membrane was removed. The kidney cell suspensions were further digested with HBSS supplemented with collagenase, pronase, and DNase for 10 min in a 37 °C water bath. The cells were then washed and centrifuged in a 13.4% Nycodenz gradient at 1400× *g* for 20 min without brake. The interface containing the enriched RSCs was collected and washed with HBSS. Then, the isolated RSCs were cultured in DMEM supplemented with 10% FBS. The purity of RSCs was assessed by microscopic observation. After reaching confluence in primary culture, the RSCs were passaged and used as activated RSCs.

### 2.3. Reverse Phase HPLC (RP-HPLC) Quantitation of Retinoids

Cells were quantified and extracted as described previously [23]. RP-HPLC was carried out on an AKTA Explorer HPLC system (GE Healthcare Life Sciences, Piscataway, NJ, USA). The chromatographic conditions were the same as previously described [24].

### 2.4. Quantitative Real-Time PCR

Total RNA was isolated using TRIzol (Ambion, Austin, TX, USA), and was used to synthesize the cDNA. Real-time PCR was performed on an ABI QuantStudio^TM^ 3 Real-Time PCR system. To control for variations in the reaction, the PCR products were normalized against the mRNA levels of glyceraldehyde 3-phosphate dehydrogenase (*GAPDH*). The primers used are listed in Appendix A.

### 2.5. Western Blotting

Cell lysates were prepared for analyses by electrophoresis and immunoblotting as described previously [22]. The primary antibodies used are listed in Appendix A.

### 2.6. Immunofluorescence

RSCs were seeded onto glass coverslips coated with gelatin. The cells were fixed with 4% paraformaldehyde and permeabilized with PBS containing 0.1% Triton X-100. The samples were then incubated overnight at 4 °C with primary antibody diluted in 1% BSA in PBST (PBS + 0.1% Tween 20), followed by incubation with FITC-, Texas Red- or Alexa Fluor^®^ 594-conjugated secondary antibody. The following primary antibodies were used: anti-albumin (Affinity Bioreagents, Rockford, IL), anti-α-SMA (Sigma-Aldrich), and anti-cytoglobin/stellate cell activation-associated protein (Cygb/STAP) (generous gift of Dr. Norifumi Kawada) [25]. After staining with 4′,6-diamidino-2-phenylindole (DAPI), RSCs were observed under a Leica TCS SP8 microscope.

### 2.7. Animal Experiments

Eight-week-old male C57BL/6 mice were used for unilateral ureteral obstruction (UUO). Left kidney UUO was performed in mice (*n* = 18) with an established protocol, and the sham-operated group (*n* = 6) was subjected to the same surgical procedure without ureter ligation. For determining the therapeutic effects of R-III, UUO-treated mice were randomly divided into two groups and injected via the tail vein with saline or R-III (30 μg) every day for seven days, starting on day eight after UUO. Mice were euthanized 15 days post-surgery, and the kidney tissues were weighed and collected for analyses. All experiments were performed in duplicate.

### 2.8. Immunohistochemistry

Formalin-fixed, paraffin-embedded kidney tissues were sectioned (5-μm thick) and stained with PAS for histological analysis or MT for analysis of collagen deposition. Tissue sections were also subjected to immunohistochemistry with the antibodies listed in Appendix A, and 10 random fields were graded semiquantitatively, and the average score was calculated as described previously [26]. All histological examinations were performed by a trained pathologist in a blinded manner. The degree of tubulointerstitial fibrosis and injury was determined by assessing the MT-stained sections. Briefly, images were captured by digital imaging sequentially over the entire sagittal section incorporating the cortex and outer medulla (10–20 images), and each image was divided into a grid of 252 squares and scored for the presence of tubular injury. The final score was the percentage of squares in each image with a positive score, averaged for all images from a single kidney.

### 2.9. Double Immunostaining

Double immunostaining was performed on Ventana Benchmark XT immunostainer (Ventana Medical System, Tucson, AZ, USA) following the manufacturer’s instructions. Briefly, for His-tag:α-SMA and His-tag:Cygb/STAP, samples immobilized on slides were incubated with His-tag antibody (Bio-Rad, Hercules, CA, USA). The bound primary antibody was detected with the OptiView DAB IHC Detection Kit (Ventana Medical System). Subsequently, the samples were incubated with α-SMA or Cygb/STAP antibody, and the bound primary antibody was detected using the ultraView Universal AP Red Detection Kit (Ventana Medical System). For His-tag:CD31, His-tag:desmin, and His-tag:F4/80, the samples were incubated with an antibody against CD31, desmin, or F4/80, respectively, and the bound primary antibody was detected with the OptiView DAB IHC Detection Kit. Subsequently, the samples were incubated with His-tag antibody and the bound primary antibody was detected by the ultraView Universal AP Red Detection Kit. The samples were then counterstained with hematoxylin (Ventana Medical Systems).

### 2.10. Statistical Analysis

Results are expressed as the means ± standard deviation (SD). A paired *t*-test was used for analysis of in vitro data. Data from in vivo studies were analyzed by either the Wilcoxon rank-sum test or the Kruskal‒Wallis test, followed by Dwass‒Steel‒Critchlow‒Fligner (DSCF) multiple comparison test. Statistical analyses were performed using SPSS for Windows, version 20.0 (SPSS, Inc., Chicago, IL, USA). A *p*-value < 0.05 was considered significant.

## 3. Results

### 3.1. Cells Resembling HSCs Are Present in the Kidney Tissues

Previous studies have suggested that vitamin A-storing cells are also present in the kidneys [15] and demonstrated that both Cygb/STAP and cellular retinol-binding protein 1 (CRBP-1), which are markers of HSCs, are expressed in the renal interstitium [27,28]. These findings prompted us investigate the possibility that renal counterparts of HSCs may exist. We isolated cells from the rat kidney tissues using the HSC isolation protocol with minor modifications and compared their morphological and biochemical characteristics with those of HSCs. 1~3 × 10^5^ cells (referred to as renal stellate cells, RSCs) were isolated per rat and seeded into culture dishes. Results showed that, like HSCs on day three after seeding (HSCs d3; early-activated cells), RSCs on day three after seeding (RSCs d3) exhibited a flattened polygonal shape and contained lipid droplets in the cytoplasm, as assessed by transmission electron microscopy and oil red O staining (Figure 1A,B, Appendix A). Their cell size and lipid droplet size were, however, smaller compared with those of HSCs d3. In addition, similar to HSCs, which when cultured on plastic undergo spontaneous transdifferentiation/activation in vitro [10], RSCs on day seven after seeding (RSCs d7) and RSCs after passage two (RSCs P2) transformed into myofibroblast-like cells and displayed loss of lipid droplets and increased expression of collagen type I and α-SMA, both well-known markers for activated HSCs/myofibroblasts (Figure 1B–D, and Figure 2A). Next, reverse-phase HPLC analysis revealed that RSCs contained retinol (vitamin A) whose level was decreased in RSCs d7 compared to that in RSCs d3, whereas the level of all-trans retinoic acid (ATRA), a bioactive vitamin A metabolite, increased with time in culture (Figure 1E,F). The observed changes in the retinoid levels in RSCs are in agreement with previous findings for HSCs [24,29], although the retinoid levels were lower in RSCs than in HSCs. Furthermore, real-time PCR analysis showed that, as observed in HSCs, the mRNA expression levels of markers of quiescent HSCs, such as PPARγ, C/EBPα, angiogenic factor with G patch and FHA domains 1 (Aggf1), and albumin [19,30,31,32], were decreased in RSCs d7 compared with those in RSCs d3 (Figure 1G,H). These finding suggest that the cells, which we isolated from the kidney tissues, are renal counterparts of HSCs.

### 3.2. Albumin Expression and R-III Treatment Inhibited the Transdifferentiation/Activation of RSCs In Vitro

Confocal microscopy showed that RSCs d3 (early-activated cells), but not RSCs P2 (activated/myofibroblastic cells), expressed albumin and emanated autofluorescence of vitamin A. On the other hand, α-SMA expression was detected in RSCs P2 (Figure 2A). Intriguingly, when RSCs P2 were transfected with a plasmid encoding albumin, they transformed into distinct cell types with autofluorescent lipid droplets reappearing and reduced α-SMA expression, as observed in HSCs [19]. Treatment of RSCs P2 with R-III also induced a phenotypic reversion into fat-storing cells and decreased the mRNA expression levels of α-SMA and collagen type I (Figure 2B,C). However, treatment with bovine serum albumin, which scarcely enters RSCs, exerted no significant effect on α-SMA expression (Appendix A). Next, we analysed the expression of other SC markers in RSCs. Western blotting revealed that lecithin retinol acyltransferase (LRAT), an enzyme responsible for retinyl ester formation and a marker for quiescent HSCs [33], was expressed in freshly isolated RSCs, but not in RSCs P2 (Figure 2D). Immunofluorescence and real-time PCR analyses also confirmed the expression of the HSC markers Cygb/STAP and CRBP-1, respectively (Figure 2E,F) [25,34]. CRBP1 expression was not detected in L6 rat myoblasts, which was used as a negative control. Thus, our findings suggest that albumin and its derivative R-III inactivates RSCs.

### 3.3. R-III Administration Reduced UUO-Induced Renal Fibrosis

Based on the in vitro effects of R-III on RSCs, we evaluated its therapeutic effects on renal fibrosis. UUO was performed in C57BL/6J mice and R-III was injected via the tail vein daily for seven days, starting on day eight after UUO (Appendix A). Histological analysis by Masson’s trichrome (MT) and periodic acid–Schiff (PAS) staining revealed that UUO induced interstitial fibrosis and tubular atrophy, which was significantly reduced by R-III treatment (Figure 3A). The histopathologic degree of tubulointerstitial fibrosis was scored in MT-stained sections, and we found that R-III administration reduced fibrosis by ~55% (Figure 3A, right panel). Next, we performed immunohistochemical staining to visualize the pro-fibrotic protein expression in the renal tissue. The expression of TGF-β1, plasminogen activator inhibitor 1 (PAI-1), and collagen type 1 and type IV was increased in UUO kidneys compared with that in sham kidneys; however, this effect significantly inhibited by R-III treatment (Figure 3B).

### 3.4. R-III Administration Reduced Pro-Fibrotic Marker Expression and Macrophage Infiltration in UUO Kidneys

Consistent with the immunohistochemical staining results (Figure 3B), real-time PCR revealed that the mRNA expression of TGF-β1, PAI-1, and collagen type I was increased in UUO kidneys compared with that in sham kidneys; however, this effect was suppressed by R-III administration (Figure 4A). Immunohistochemical analysis also showed that the protein expression of myofibroblast markers, such as α-SMA, desmin, and fibronectin was enhanced in UUO kidneys, but this effect was inhibited by R-III treatment (Figure 4B). Next, we assessed macrophage infiltration, as increased infiltration of macrophages has been reported in UUO kidneys [35]. Infiltration of F4/80-positive macrophages was remarkably increased in UUO kidneys compared with that in sham kidneys; however, this effect was suppressed by R-III treatment (Figure 4C). In addition, the mRNA expression level of monocyte chemoattractant protein 1 (MCP-1), a pro-inflammatory gene, also roughly paralleled the intensity of F4/80 immunostaining (Figure 4A).

### 3.5. R-III Was Delivered to RSCs In Vivo

We then investigated the cellular distribution of injected His-tagged R-III by immunohistochemistry. Unlike the sham kidneys, the UUO+R-III-treated kidneys showed distinct His-tag staining. The His-tag-positive cells were predominantly located in the peritubular and perivascular region, although a weak signal was also detected in epithelial cells (Figure 5A). Importantly, double immunostaining revealed that ~70% of Cygb/STAP^+^ cells and ~40% of α-SMA^+^ cells showed His-tag signals (Figure 5B). However, no significant overlap was found with other cellular markers, such as CD31 and F4/80, indicating that His-tagged R-III was not randomly distributed in different types of cells but successfully delivered to Cygb/STAP-positive RSCs. PDGFR-β has been shown to be expressed in myofibroblasts and renal mesenchymal cells, including mesangial cells, interstitial fibroblasts and pericytes, and vascular smooth musle cells [36]. Intriguingly, ~55% of interstitial PDGFR-β^+^ cells expressed Cygb/STAP (Figure 6C).

### 3.6. EMT Is Not Involved in the Anti-Fibrotic Effect of R-III

As EMT was previously reported to contribute to renal fibrosis [4], we examined the effects of R-III on TGF-β-induced EMT. TGF-β1 treatment caused a decrease in E-cadherin and increase in α-SMA levels in NRK-52E rat renal tubular epithelial cells, and R-III treatment partially inhibited TGF- β1-driven EMT (Appendix A). Next, we prepared cell lysates from the kidney tissues and analyzed EMT marker expression by western blotting. In line with our immunohistochemical data (Figure 4B), α-SMA protein levels were increased in UUO kidneys, but this effect was inhibited by R-III administration (Appendix A). However, we did not find any statistically significant changes in the levels of other EMT markers, such as E-cadherin and fibroblast-specific protein 1 (FSP-1), suggesting that the in vivo anti-fibrotic effect of R-III was not due to inhibition of EMT.

### 3.7. R-III Does not Affect NRK-49F Renal Fibroblasts

It has been previously suggested that myofibroblasts in the kidney represent an activated population of residual fibroblasts [37]. Hence, we examined the effect of R-III on NRK-49F rat renal fibroblasts. TGF-β1 treatment increased α-SMA mRNA levels in NRK-49 cells, which was consistent with previous findings [38], but R-III treatment had no significant effect (Figure 6A). Thus, it is unlikely that the in vivo anti-fibrotic effect of R-III is derived from its action on resident renal fibroblasts.

### 3.8. STRA6 Is Expressed in the Kidneys

As the entry of R-III into HSCs is dependent on STAR6, a specific membrane receptor for RBP [21,22], the expression of STRA6 was examined in the kidney. Immunohistochemistry showed that STRA6 staining was mainly detected in the peritubular and perivascular cells, resembling His-tag staining (Figure 5A), and significantly increased in UUO kidneys compared with that in sham kidneys (Figure 6B). Real-time PCR analysis revealed that STRA6 mRNA expression levels roughly paralleled the intensity of STRA6 immunostaining (Figure 6C).

### 3.9. Lipid Droplet-Containing Cells Appeared in R-III-Treated UUO Kidney

As R-III treatment induced phenotypic reversion of RSCs P2 (activated/myofibroblastic cells) into lipid droplet-containing cells in vitro (Figure 2B), we examined whether the reappearance of fat-storing cells also occurs in vivo. Consistent with previous findings [15], electron microscopy data revealed that lipid droplet-containing cells were not present in UUO kidneys (Figure 6D). Lipid droplet-containing cells were, however, detected in the peritubular region of the UUO+R-III-treated kidneys.

### 3.10. RSCs Express Pericyte Markers

HSCs are pericytes found in the perisinusoidal space of the liver [9]. To verify the identity of our isolated RSCs, we examined the expression of the commonly used pericyte markers, including α-SMA, PDGFR-β, and chondroitin sulfate proteoglycan neural/glial antigen 2 (NG2) [39], by real-time PCR. Results showed that the mRNA expression levels of α-SMA, PDGFR-β, and NG2 in RSCs were increased during cell culture (Figure 6E). However, the NG2 mRNA level was decreased in HSCs d7 (Figure 6F). Notably, the mRNA levels of collagen type I and α-SMA in RSCs d7 (activated cells) were >1000-fold higher when compared to those in RSCs d1 (pre-activated cells). These results suggest that our isolated cells from the kidney tissues may be, at least in part, renal pericytes and that the in vivo anti-fibrotic effect of R-III is probably due to its action on RSCs.

## 4. Discussion

Hepatic and pancreatic stellate cells were first isolated and characterized from the rat liver and pancreas in mid 1980s and late 1990s, respectively [40,41]. In this study, we isolated stellate cells from rat kidneys using the standard protocol for HSC isolation and characterized them. We also tried to isolate RSCs from mouse kidneys but failed to obtain sufficient cells for analysis, probably due to either technical reasons or a low quantify of cell population or both. (Similarly, it is much more difficult to isolate mouse HSCs than rat HSCs.) RSCs displayed morphological and biochemical characteristics similar to those of HSCs. RSCs d1 (pre-activated) or d3 (early-activated) contained autofluorescent, cytoplasmic lipid droplets and showed little or no α-SMA expression, as has been observed in HSCs and PSCs [10,13]. By contrast, culture-activated RSCs d7, P1, or P2 lost most of their lipid droplets and displayed a myofibroblastic phenotype with increased expression of α-SMA and collagen type I. Such a transition into myofibroblasts during cell culture resembles the activation process of HSCs and PSCs. Furthermore, RSCs were found to contain Vitamin A and express the stellate cell markers, such as LRAT, Cygb/STAP, and CRBP-1. These findings indicate that our isolated cells are renal counterparts of HSCs/PSCs and differ from fibroblasts.

Previous studies have suggested that scar-producing myofibroblasts following kidney injury are mainly derived from pericytes and perivascular fibroblasts, based on their perivascular localization and expression of PDGFR-β [6,7]. A question then arises; are the isolated RSCs are renal pericytes? Our study showed that RSCs express common pericyte markers, PDGFR-β and NG2, and that Cygb/STAP (or His-tag)-positive cells are located in perivascular region. This suggests that RSCs may be, at least in part, renal pericytes as HSCs are liver pericytes, although clear evidence is lacking. Further study is required to address this issue, but there is a serious technical problem that there are no specific markers for stellate cells and pericytes. Commonly used markers, including PDGFR-β, NG2, and Cygb/STAP, are expressed in different types of interstitial, mesenchymal stromal cells [36,42,43].

Consistent with a previous finding that the expression of NG2 was increased in pericytes/coll1a1^+^ cells after UUO [6], we found that the mRNA expression of NG2 in RSCs was increased during culture activation. By contrast, NG2 expression was decreased in HSCs d7 (activated HSCs) compared to that in HSCs d1. The mechanism underlying the differential regulation of NG2 expression and its physiological significance are not yet clear and require further investigation.

Despite intensive research efforts over the last 30 years, the molecular mechanism of HSC/PSC activation remains elusive. Quiescent HSCs store retinoids in the form of retinyl esters in the cytoplasmic lipid droplets. When the lipid droplets are rapidly lost during SC activation, a portion of their retinoid content is likely released and metabolized into retinaldehyde, which is further irreversibly oxidized to retinoic acid (RA) by retinaldehyde dehydrogenase. A role of retinoids in HSC activation has been proposed, but previous reports regarding the effects of exogenous retinoids on HSC activation and liver fibrosis are controversial [44]. Recently, we demonstrated that endogenous RA plays a role in HSC activation [22], which was in agreement with a previous study that showed that the expression of aldehyde dehydrogenase 1 family member A2, also known as retinaldehyde dehydrogenase 2, was strikingly upregulated in myofibroblasts in UUO kidney compared to that in their precursors [45]. Biophysical studies have shown that albumin readily binds to RA [18,46]. This explains why albumin expression or R-III treatment markedly reduces intracellular RA levels and RA signaling in activated HSCs [22]. The mechanism of action of R-III downstream of RA sequestration is now under investigation.

In this study, we showed that R-III treatment induces phenotypic reversion of activated/myofibroblastic RSCs into fat-storing phenotype but exerts no significant effects on renal fibroblasts. In addition, R-III administration led to the reappearance of lipid droplet-containing cells in R-III-treated kidneys. Thus, it is reasonable to assume that the in vivo anti-fibrotic effect of R-III is due to its action on RSCs. The UUO mouse model was used because (1) this significantly reduced the amount of R-III needed for these critical in vivo experiments and (2) a recent Meta-analysis study of the UUO model suggests that both mouse and rat models equally contribute to the study of renal fibrosis [47].

The stellate cells of the liver and extrahepatic organs, such as the pancreas and kidneys, show striking similarities with respect to morphology and perivascular location, which suggests that their activation contributes to the myofibroblast population present in fibrotic tissues [13]. There is an imperative demand for the development of tissue fibrosis therapy and several studies have been previously conducted with an aim to inactivate SCs and reduce tissue fibrosis, but unfortunately an effective therapy for fibrosis is not yet available.

## 5. Conclusions

We demonstrated that our isolated RSCs from the kidney tissues are the renal counterparts of HSCs, and that R-III exerts anti-fibrotic effects on RSC transdifferentiation/activation and renal fibrosis, similar to its effects on HSC activation and liver fibrosis. Hence, our findings suggest that R-III, designed for stellate cell targeting, may serve as a novel anti-fibrotic drug candidate.

## Figures and Tables

**Figure 1 biomedicines-08-00431-f001:**
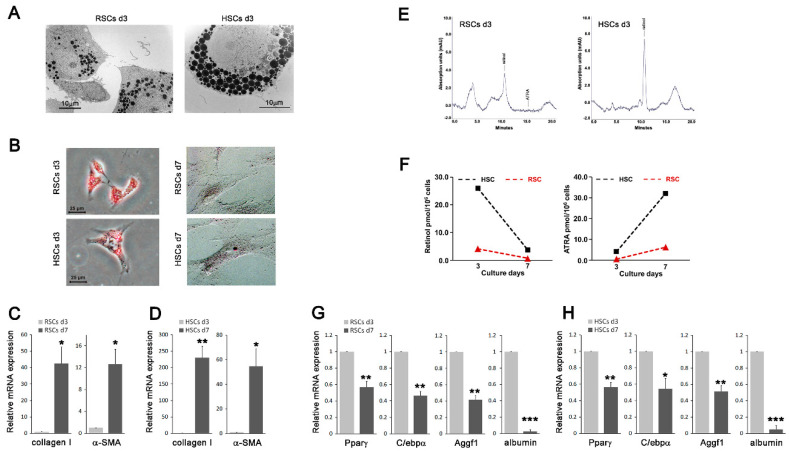
Renal stellate cells (RSCs) share morphological and biochemical characteristics with hepatic stellate cells (HSCs). (**A**) Transmission electron microscopy showing the presence of cytoplasmic lipid droplets in RSCs on day 3 after seeding (RSCs d3) and HSCs d3. (**B**) RSCs and HSCs on days 3 and 7 after seeding were subjected to oil red O staining. Representative images from each group are shown. (**C**,**D**,**G**,**H**) Total RNA was isolated from the RSCs (**C**,**G**) and HSCs (**D**,**H**) on days 3 and 7 after seeding, and the expression levels of collagen type I, α-SMA (**C**,**D**), PPARγ, C/EBPα, Aggf1 and albumin (**G**,**H**) were analyzed by real-time PCR. The data represent the means ± SD for three independent experiments. *p*-value was calculated using paired *t*-test (compared with cells on day 3 after seeding). * *p* < 0.05, ** *p* < 0.01, *** *p* < 0.001. (**E**) Retinol levels in whole-cell extracts of RSCs d3 and HSCs d3 were measured by reverse-phase HPLC. (**F**) Graphical representation of intracellular retinol and ATRA levels in RSCs and HSCs on days 3 and 7 after seeding.

**Figure 2 biomedicines-08-00431-f002:**
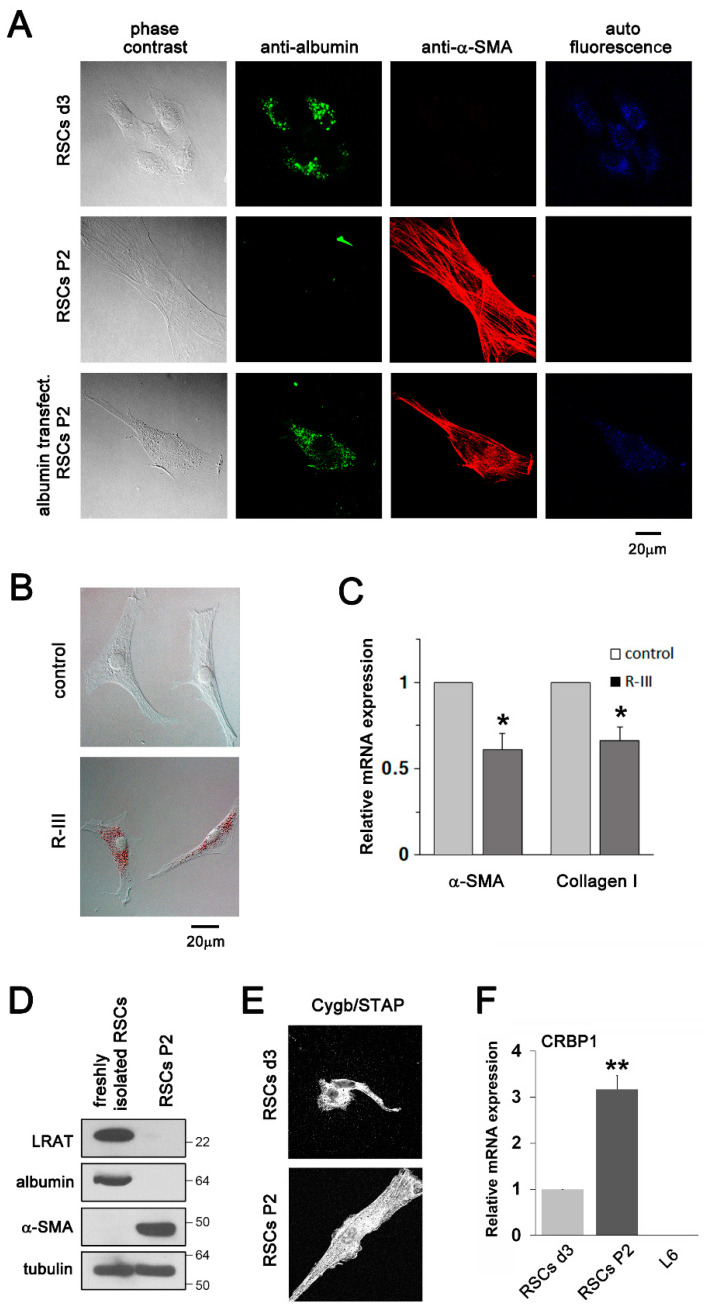
Albumin and its derivative R-III inhibit the activation of renal stellate cells (RSCs) in vitro. (**A**) Phase contrast images, immunofluorescence images using antibodies against albumin or α-SMA, and autofluorescence images are shown for RSCs on day 3 after seeding (RSCs d3), RSCs after passage 2 (RSCs P2), and RSCs P2 transfected with a plasmid encoding albumin, respectively. (**B**,**C**) RSCs P2 were treated with or without HPLC-purified R-III (0.5 μM) for 20 h and analyzed by oil red O staining (**B**) and real-time PCR for the expression of α-SMA and collagen type I (**C**). (**D**) Cell lysates from freshly isolated RSCs and RSCs P2 were analyzed by western blotting. α-tubulin was used as a loading control. Full-length blots are presented in Appendix A. (**E**) RSCs d3 and RSCs P2 were analyzed by immunofluorescence using anti-Cygb/STAP antibody. (**F**) Total RNA was extracted from RSCs d3, RSCs P2, and L6 rat myoblasts, and the CRBP-1 expression was analyzed by real-time PCR. The data represent the means ± SD for three independent experiments. *p*-value was calculated using paired *t*-test (*n* = 3). * *p* < 0.05, ** *p* < 0.01.

**Figure 3 biomedicines-08-00431-f003:**
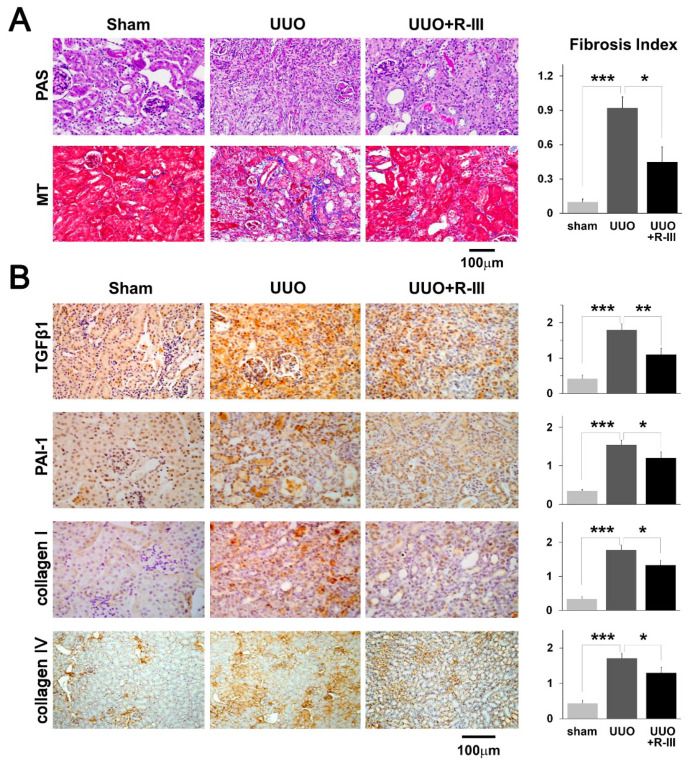
R-III reduces unilateral ureteral obstruction (UUO)-induced renal fibrosis. (**A**) Kidney sections from sham-, UUO-, and UUO+R-III-treated mice were stained with PAS and MT. The right panel shows the semiquantitative score of tubulointerstitial fibrosis. (**B**) Kidney sections from sham- and UUO- and UUO+R-III-treated mice were subjected to immunohistochemistry against TGF-β1, PAI-1, and collagen type I and type IV. Representative images from each study group are shown. Semiquantitative analysis of the staining intensity for each group is shown (right). Data are expressed as the means ± SD (*p*-value; Kruskal‒Wallis test, followed by DSCF multiple comparison test). * *p* < 0.05, ** *p* < 0.01, *** *p* < 0.001.

**Figure 4 biomedicines-08-00431-f004:**
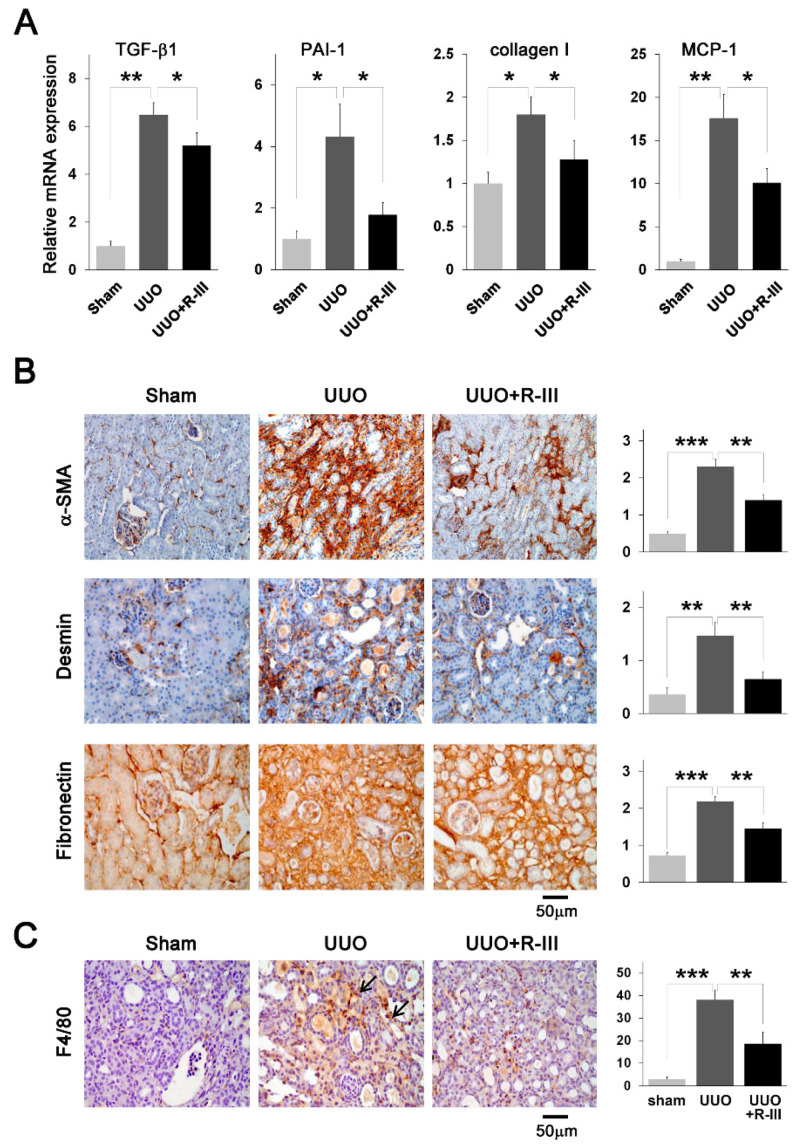
R-III administration reduces the expression of the pro-fibrotic and pro-inflammatory genes in UUO kidneys. (**A**) Total RNA was extracted from the kidneys of sham-, UUO-, and UUO+R-III-treated mice for the analysis of TGF-β1, PAI-1, collagen type I, and MCP-1 expression by real-time PCR. Data are expressed as the means ± SD (*p*-value; Kruskal‒Wallis test, followed by DSCF multiple comparison test). * *p* < 0.05, ** *p* < 0.01, *** *p* < 0.001 (**B**) Kidney sections from sham-, UUO-, and UUO+R-III-treated mice were subjected to immunohistochemistry against α-SMA, desmin, and fibronectin. Representative images from each study group are shown. Semiquantitative analysis of the staining intensity in each group is shown (right). (**C**) The kidney sections from treated mice were subjected to immunohistochemistry against F4/80. Arrows indicate F4/80-positive cells. The number of F4/80-positive cells was quantified (right).

**Figure 5 biomedicines-08-00431-f005:**
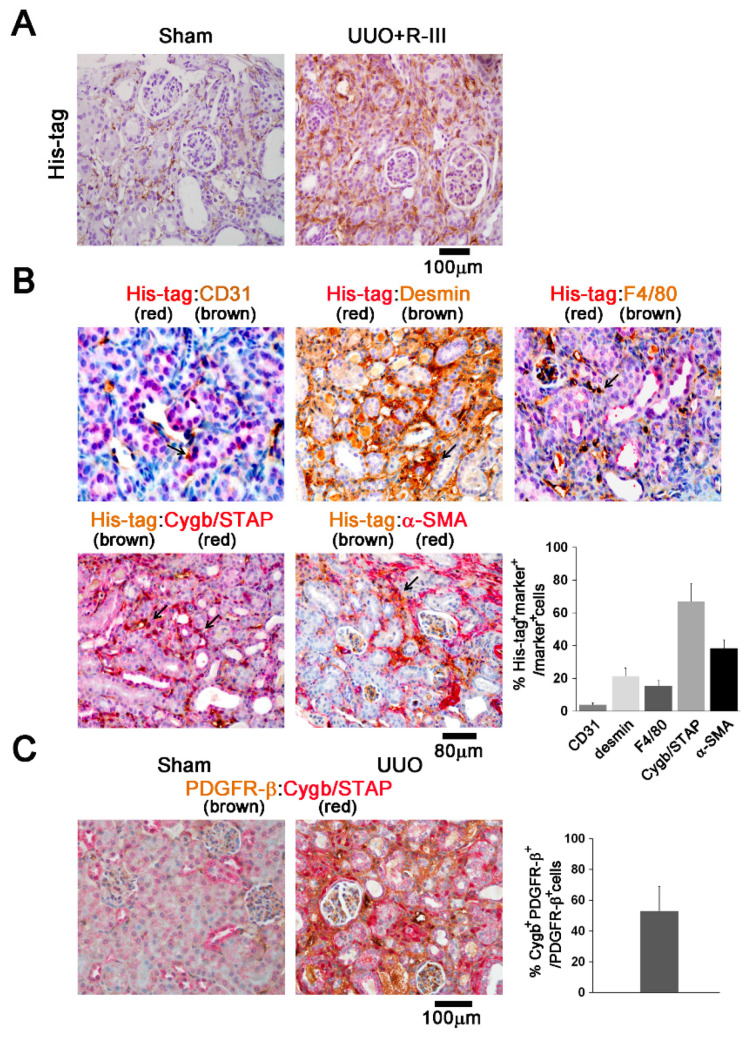
Injected R-III is delivered to RSCs in vivo. (**A**) Kidney sections from sham- and UUO+R-III-treated mice were subjected to immunohistochemistry against His-tag. (**B**) Double immunohistochemistry was performed for kidney sections from UUO+R-III-treated mice. For His-tag:CD31, His-tag:desmin, and His-tag:F4/80, His-tag antibody stains red, and antibodies against CD31, desmin, and F4/80 stain brown. For His-tag:Cygb/STAP and His-tag:α-SMA, His-tag antibody stains brown, and antibodies against Cygb/STAP and α-SMA stain red. His-tag staining largely overlaps with Cygb/STAP staining. Arrows indicate co-localization of His-tag and selected markers. Semiquantitative scoring of His-tag^+^ selected marker^+^ double-positive cells as a percentage of selected marker-stained cells is shown on the right. Data are expressed as the means ± SD (**C**) Double immunohistochemistry was performed for kidney sections from UUO-treated mice. For PDGFR-β:Cygb/STAP, PDGFR-β antibody stains brown, and Cygb/STAP antibody stains red.

**Figure 6 biomedicines-08-00431-f006:**
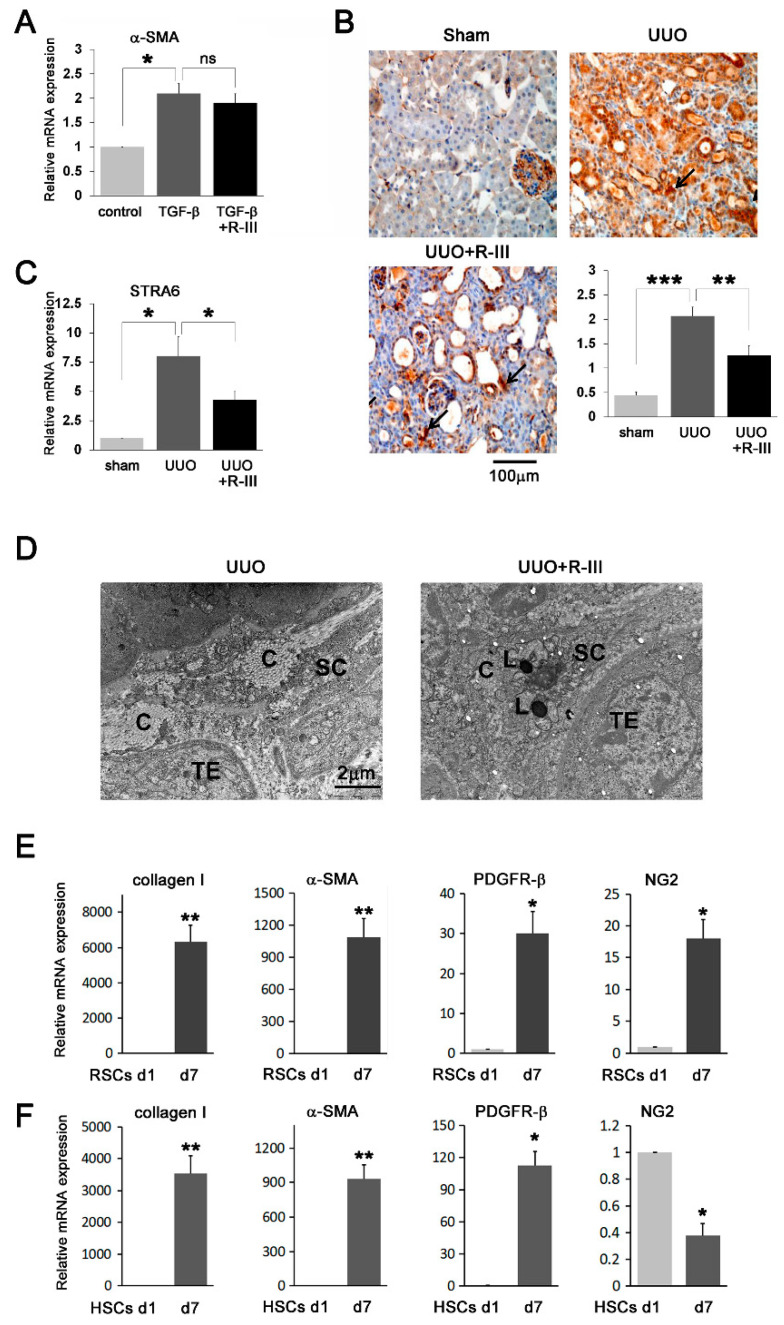
RSCs express pericyte markers. (**A**) NRK-49F rat kidney fibroblast cells were treated with TGF-β1 (50 ng/mL) in the presence or absence of R-III (0.5 μM), and α-SMA expression was analyzed by real-time PCR. The data represent the means ± SD for three independent experiments. *p*-value was calculated using paired t-test (*n* = 3). * *p* < 0.05. (**B**) Kidney sections from sham-, UUO-, and UUO+R-III-treated mice were subjected to immunohistochemistry against STRA6. Arrows indicate STRA6-positive cells. Semiquantitative analysis of STRA6 staining is shown (right). Data are expressed as the means ± SD (*p*-value; Kruskal‒Wallis test, followed by DSCF multiple comparison test). ** *p* < 0.01, *** *p* < 0.001. (**C**) Total RNA was extracted from the kidneys of sham-, UUO-, and UUO+R-III-treated mice, and STRA6 expression was analyzed by real-time PCR. (**D**) Transmission electron microscopy showing the lipid droplet-containing cells in the peritubular region of UUO+R-III-treated kidneys. SC, stellate cells; TE, tubular epithelial cells; L, lipid; C, collagen fibers. (**E**,**F**) Total RNA was isolated from RSCs (E) and HSCs (F) on days 1 and 7 after seeding, and the expression of collagen type I, α-SMA, PDGFR-β, and NG2 was analyzed by real-time PCR. *p*-value was calculated using paired t-test (*n* = 3, compared with cells on day 1 after seeding). * *p* < 0.05, ** *p* < 0.01.

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
