# Peer review of "Inhibition of Renal Stellate Cell Activation Reduces Renal Fibrosis"

_biomedicines, 2020, doi:10.3390/biomedicines8100431_

Round 1
Reviewer 1 Report
This is an intersiting study revealing a hepatic stella cell like cell population in kidney, supposed to be the pericyte, and successive targeting this population of cells by a recombinant fusion protein (designated R-III) (domain III of
82 albumin was fused to the C-terminus of retinol-binding protein ) reduced kidney fibrosis in UUO model, an anti-fibrotic effect authors previously tested in hepatic fibrosis.
This work is novel, and has translational potential for treatment of kidney fibrosis
- while in vitro cell experiments were all using cells isolated from rat, and rat cell lines, UUO model was performed in mouse. can authors explain the reason why UUO in rat was not used.
- To be more easy to translate into renal fields, would it be more appropriate to name RSC as SC like pericyte or simply pericyte.
- Whether evidence to prove RSC are pericyte are better provided
- Fig 4 & 5 desmin staining meant to stain for Podocyte? If so, not many glomerulars were presented in images
- What is the interpretation for results Fig 5B quantitation data of His: CD31, His;F4/80 and His: desmin double positive cells.
- Would be better for readers to follow, if Fig5 double staining to be annotated with colour /marker name.
Author Response
I am grateful for the comments and suggestions of the reviewers to improve our manuscript. Here is a point-by-point response to the reviewer's comments.
1. while in vitro cell experiments were all using cells isolated from rat, and rat cell lines, UUO model was performed in mouse. can authors explain the reason why UUO in rat was not used.
→ Our current supply of highly purified, recombinant R-III reagent is limited and our current laboratory staffing is restricted due to COVID-19 precautions. Hence rapid production of additional reagents is not easily performed. A recent Meta-analysis study of the UUO model in both rat and mouse models (Matínez-Kilmova et al., Biomolecules 2019, 8(4):141. doi: 10.3390/biom9040141) suggests that these models contribute equally to the study of renal fibrosis. We used the UUO mouse model because this significantly reduced the amount of albumin-RBP fusion protein (R-III) needed for these critical in vivo experiments. We included this in the discussion of the revised manuscript (page 15, line 443).
2. To be more easy to translate into renal fields, would it be more appropriate to name RSC as SC like pericyte or simply pericyte.
→ In this study, we focus significant effort on demonstrating that our isolated cells are renal counterparts of hepatic stellate cells although our data suggest that these cells may possibly be pericytes. However, this characterization is based solely on their perivascular localization and the expression of few pericyte markers. Thus in order to prevent overstating our conclusions, we prefer to refer to these cells as RSCs, until we are able to conduct more extensive characterization to include more direct evidence of their definitive identification as pericytes. We stated that such definitive evidence was lacking in our current manuscript (as mentioned in the discussion).
3. Whether evidence to prove RSC are pericyte are better provided
→ See the above answer to reviewer query #2. Future studies will specifically determine the identity of these RSC cells utilizing immunohistochemical, immunofluorescent methods and/or flow cytometry methodology.
4. Fig 4 & 5 desmin staining meant to stain for Podocyte? If so, not many glomerulars were presented in images
→ As UUO is characterized by an increase in interstitial fibrosis and the immunohistochemical staining for desmin, a marker of myofibroblasts, is used as an index of interstitial fibrosis, desmin staining was shown in Figure 4B. On the other hand, the reason why desmin staining did not show substantial overlap with His-tag staining in double immunohistochemistry is not clear at present. However, based on your suggestion we will in future studies examine the colocalization of the His-tag R-III with mesangial cell staining.
5. What is the interpretation for results Fig 5B quantitation data of His: CD31, His;F4/80 and His: desmin double positive cells.
→ In our previous studies, we have shown that R-III (albumin-RBP fusion protein) enters hepatic stellate cells in a RBP receptor-dependent manner. No significant overlap with CD31 and F4/80 indicates that His-tagged R-III was not randomly distributed in different types of cells. We have reworded the sentence (page11 line 312) in the revised manuscript.
6. Would be better for readers to follow, if Fig5 double staining to be annotated with colour /marker name.
→ We have revised the figure #5 as suggested (page 11).
Reviewer 2 Report
Inhibition of renal stellate cells with R-III has shown anti-renal fibrotic effects in UUO model. The study design is well-organized and the results were convincing. Especially, the authors identified renal stellate cell and confirmed its relevance to renal fibrosis and R-III had beneficial effects on UUO-induced renal fibrosis. I have no concrens regarding to thiis article.
Author Response
I am grateful for the comments of the reviewer.